# Diagnostic Performance of Contrast-Enhanced Digital Mammography versus Conventional Imaging in Women with Dense Breasts

**DOI:** 10.3390/diagnostics13152520

**Published:** 2023-07-28

**Authors:** Giuliana Moffa, Francesca Galati, Roberto Maroncelli, Veronica Rizzo, Federica Cicciarelli, Marcella Pasculli, Federica Pediconi

**Affiliations:** Department of Radiological, Oncological and Pathological Sciences, Sapienza University of Rome, 00161 Rome, Italy; francesca.galati@uniroma1.it (F.G.); roberto.maroncelli@uniroma1.it (R.M.); veronica.rizzo@uniroma1.it (V.R.); federica.cicciarelli@uniroma1.it (F.C.); marcella.pasculli@uniroma1.it (M.P.); federica.pediconi@uniroma1.it (F.P.)

**Keywords:** breast density, digital mammography, breast ultrasound, contrast-enhanced mammography, diagnosis

## Abstract

The aim of this prospective study was to compare the diagnostic performance of contrast-enhanced mammography (CEM) versus digital mammography (DM) combined with breast ultrasound (BUS) in women with dense breasts. Between March 2021 and February 2022, patients eligible for CEM with the breast composition category ACR BI-RADS c–d at DM and an abnormal finding (BI-RADS 3-4-5) at DM and/or BUS were considered. During CEM, a nonionic iodinated contrast agent (Iohexol 350 mg I/mL, 1.5 mL/kg) was power-injected intravenously. Images were evaluated independently by two breast radiologists. Findings classified as BI-RADS 1–3 were considered benign, while BI-RADS 4–5 were considered malignant. In case of discrepancies, the higher category was considered for DM+BUS. Sensitivity, specificity, positive predictive value (PPV), negative predictive value (NPV), and accuracy were calculated, using histology/≥12-month follow-up as gold standards. In total, 51 patients with 65 breast lesions were included. 59 (90.7%) abnormal findings were detected at DM+BUS, and 65 (100%) at CEM. The inter-reader agreement was excellent (Cohen’s k = 0.87 for DM+BUS and 0.97 for CEM). CEM showed a 93.5% sensitivity (vs. 90.3% for DM+BUS), a 79.4–82.4% specificity (vs. 32.4–35.5% for DM+BUS) (McNemar *p* = 0.006), a 80.6–82.9% PPV (vs. 54.9–56.0% for DM+BUS), a 93.1–93.3% NPV (vs. 78.6–80.0% for DM+BUS), and a 86.1–87.7% accuracy (vs. 60.0–61.5% for DM+BUS). The AUC was higher for CEM than for DM+BUS (0.865 vs. 0.613 for Reader 1, and 0.880 vs. 0.628, for Reader 2) (*p* < 0.001). In conclusion, CEM had a better diagnostic performance than DM and BUS alone and combined together in patients with dense breasts.

## 1. Introduction

Breast cancer represents a serious public health issue, being the most commonly diagnosed cancer (11.7% of total cases for both sexes) and the leading cause of cancer death in the female population [1]. Fortunately, breast cancer mortality has progressively decreased in recent decades, mainly due to improvements in treatment and early detection via massive screening campaigns [2].

Digital mammography (DM) plays a pivotal role in breast cancer imaging, both in the routine clinical practice and screening settings, enabling early detection and conservative surgery. Moreover, DM has been proven to reduce breast cancer mortality by up to over 20–30% [3]. Unfortunately, DM is burdened by limited sensitivity, especially in women with dense breasts, and relatively high false positive (FP) and false negative (FN) rates. In a recent study including approximately 112,000 examinations, Wanders at al. [4] have demonstrated a 25% reduction in sensitivity and a 13‰ increase in FP rates as the breast density increased from almost entirely fatty to extremely dense, according to the breast density categories of the 2013 American College of Radiology (ACR) Breast Imaging Reporting and Data System (BI-RADS) [5] (Figure 1).

High breast density (defined as ACR BI-RADS c–d) is associated with two main problems: it is an independent breast cancer risk factor [6] and causes a “masking effect” due to overlapping breast tissues, potentially delaying the diagnosis of non-calcified breast cancers. 

Some possible solutions have been proposed through the years in order to overcome the limits of DM. Adding digital breast tomosynthesis (DBT) to DM has the proven advantage of increasing the breast cancer detection rate both in dense and fatty breasts, while reducing recall rates, thus improving both the sensitivity and specificity of DM, with a moderate increase in the radiation dose, acquisition time and reading time [7,8,9]. Nevertheless, the available evidence suggests that DBT-detected cancers tend to be smaller, of lower grade, and of favorable histology [8], raising reasonable issues about overdiagnosis.

Breast ultrasound (BUS) is commonly used as a supplemental, low-cost diagnostic tool to increase sensitivity, especially for young women with dense breasts in whom DM and DBT have a lower sensitivity. In the screening setting, adjunctive BUS has increased the sensitivity and detection rate of early cancers and has reduced interval cancers in women with dense breasts [10,11,12,13,14,15,16], but has significantly reduced specificity and substantially increased FP rates [10,11,17]. Moreover, BUS is operator-dependent and poorly repeatable.

Considering the increasing evidence about underdiagnosis of current screening strategies, the European Society of Breast Imaging (EUSOBI) now recommends supplemental screening using contrast-enhanced magnetic resonance imaging (CE-MRI) of the breast in women with extremely dense breasts aged 50–70 years [18]. Nevertheless, the availability of breast MRI remains patchy and extremely varied in terms of equipment, staff, and experience.

Contrast-enhanced digital mammography (CEM) is a new contrast-enhanced imaging technique, first introduced in 2011, that combines the anatomic and morphologic information provided by mammography projections with functional information via intravenous iodinated contrast media administration. Similarly to breast CE-MRI, CEM is able to detect the abnormal enhancement of malignant breast lesions caused by tumoral neoangiogenesis.

CEM is performed using a standard mammography system implemented with a specific filter for the acquisition of high-energy images and a software that is able to process raw information and construct the so-called recombined images.

The main disadvantages of CEM are the increase in the radiation dose compared to DM (varying between 15 and 80% [19,20,21,22]), the use of iodinated contrast media, and the higher costs. Nevertheless, CEM is currently considered safe, since the radiation dose—although higher—remains within acceptable levels and serious adverse reactions to the contrast agent are extremely rare, accounting for about 0.002% in a recent review by Zanardo et al. including more than 14,000 patients [23]. Concerning the expenses associated with CEM, most existing digital mammography units can be implemented with dedicated filters and software to perform CEM, in order to avoid the waste of resources and reduce equipment costs.

Many authors have demonstrated that CEM outperforms DM [24,25,26,27,28,29,30], but there are only a few studies, to our knowledge, that have compared the diagnostic performance of CEM versus DM+BUS combined together [29,30,31]: one is a large population-based but retrospective study [31], and the others [29,30] have actually evaluated the diagnostic performance of CEM as an adjunct to DM (that has been associated with overestimation in tumor size [32]), or to DM+BUS.

The aim of this prospective study was to compare the diagnostic performance of CEM alone versus conventional imaging techniques combined together, in women with dense breasts.

## 2. Materials and Methods

### 2.1. Study Design and Population

The present prospective study was conducted according to the Good Clinical Practice guidelines and obtained the approval of our institutional review board.

All patients eligible for CEM examination in the diagnostic phase in the Breast Imaging Department of our Institution between March 2021 and February 2022 were considered for this study. Non-eligibility criteria included the following: suspected/ongoing pregnancy or lactation, history of allergic reaction to iodinated contrast medium, and impaired renal function, according to the European Society of Urogenital Radiology (ESUR) guidelines on contrast agents [33].

All patients participated voluntarily and provided written informed consent prior to enrollment.

The inclusion criteria were as follows: (1) breast composition category c and d at DM, according to the 2013 ACR BI-RADS [5]; (2) conventional imaging (DM and BUS) performed in the Breast Imaging Department of our Institution in the six weeks prior to CEM examination; and (3) abnormal finding classified as BI-RADS 3-4-5, according to the 2013 ACR BI-RADS, at DM and/or BUS.

The exclusion criteria were as follows: (1) inability to complete CEM; (2) previous history of breast cancer/recurrent disease; (3) ongoing neoadjuvant therapy or other cancer treatment; (4) presence of breast implants, in order to avoid the known artifact effect of implants on recombined images [34]; (5) core needle biopsy (CNB) or vacuum-assisted biopsy (VAB) performed less than 14 days prior to CEM, in order to avoid possible bias due to the procedures (e.g., post-biopsy hematoma); (6) incomplete clinical/histological data; and (7) patient’s referral to other institutions.

Data were acquired anonymously and collected using Excel 2016 (Microsoft Corp., Redmont, WA, USA).

### 2.2. Imaging Techniques

DM and CEM were performed on a low-dose digital mammography unit (Giotto Class; IMS Giotto, Bologna, Italy) capable of performing full-field 2D DM and CEM. The technical features of the system employed are summarized in Table 1.

DM included at least standard bilateral craniocaudal (CC) and mediolateral oblique (MLO) projections for all patients.

Before CEM examination, a non-ionic iodinated contrast agent (Iohexol 350 mgI/mL; Omnipaque 350, GE Healthcare, Chicago, IL, USA) was power-injected through an antecubital venous access (18–20 gauge) at a dose of 1.5 mL/kg body weight and at a rate of 3 mL/s, and followed by a 20 mL saline flush at the same rate. About 2 min after the injection, the patient was positioned for mammographic imaging, starting the scan from the breast with the suspicious finding. Two CC and two MLO views (early and late) of the affected side and of the other side were obtained sequentially within 6 min maximum (total acquisition time = 8 min). Two acquisitions (low and high energy) were performed serially for each view. A post-processing-dedicated software was used to construct the recombined images showing areas of post-contrast enhancement.

Imaging of pre-menopausal or peri-menopausal women was performed between the 7th and the 14th day of the menstrual cycle in order to minimize the effects of background parenchymal enhancement on the examination accuracy.

BUS examination was performed by two experienced breast radiologists using a high-resolution ultrasound unit with a 12 MHz linear probe (Affiniti 70 G; Philips, Amsterdam, The Netherlands).

### 2.3. Imaging Evaluation and Interpretation

All imaging evaluation was performed independently by two experienced breast radiologists (Reader 1: F.G. and Reader 2: G.M., with 10 and 5 years of experience, respectively). DM and CEM evaluation was performed at a dedicated workstation equipped with an integrated software (Raffaello, IMS Giotto, Bologna, Italy) and two dedicated 5-Megapixel diagnostic LED monitors (GX570; Eizo, Hakusan, Japan). The readers were aware of the aim of the study but were blinded to the results of previous breast examinations and of clinical and histopathological information.

Conventional imaging assessment (DM followed by BUS) was performed using all images available and suspicious findings were identified and classified following the 2013 ACR BI-RADS lexicon for DM [5] and ultrasound [35]. Breast composition was evaluated at DM and classified visually as ACR BI-RADS c in case of heterogeneous breast density and as ACR BI-RADS d in case of extreme density [5].

The presence of calcifications at DM was annotated.

CEM imaging assessment was performed 1–2 weeks after conventional imaging in order to avoid possible biases, using both low-energy and recombined images. Findings were classified according to the recently introduced CEM supplement to the 2013 ACR BI-RADS Mammography [36].

A final BI-RADS assessment category was provided for each finding at all imaging techniques. In the event of discrepancies between the DM and BUS assessment category, the higher BI-RADS category was taken into consideration for further analysis.

To evaluate the diagnostic performance of DM+BUS and CEM, findings classified as BI-RADS 1–3 were considered benign, while findings classified as BI-RADS 4–5 were considered malignant. A BI-RADS category 0 was not allowed.

The histopathological results from CNB, VAB or surgery (when available) or a ≥12-month long follow-up period were considered as the reference standard.

### 2.4. Statistical Analysis

The statistical analyses and the graph plotting were performed using IBM SPSS Statistics 27.0 (IBM, Armonk, NY, USA).

Sensitivity, specificity, positive predictive value (PPV), negative predictive value (NPV), and accuracy were calculated for DM, BUS, DM+BUS, and CEM.

The receiver operating characteristic (ROC) curve and the corresponding area under the curve (AUC) value were used to estimate the overall diagnostic performance of all imaging techniques.

The McNemar test was used to compare sensitivities and specificities of DM+BUS vs. CEM.

Each analysis was performed separately for Reader 1 and Reader 2. Inter-reader agreement was calculated using Cohen’s k coefficient.

*p*-values < 0.05 were considered statistically significant.

## 3. Results

In total, 110 CEM examinations were performed in the Breast Imaging Department of our Institution between March 2021 and February 2022. Among these, 63 patients met the inclusion criteria. In two cases, the examination was interrupted before being completed, 7 patients had previous history of breast cancer/suspected recurrency, 1 patient had breast implants, and 2 patients had been referred to another institution. Therefore, 51 patients with 65 breast lesions were included in the study.

The patients’ age ranged from 40 to 81 years (median age = 55.5, mean age = 57.8, SD = 10.2).

The breast composition at DM was classified as ACR BI-RADS c in 92.2% and ACR BI-RADS d in 7.8% of cases, respectively. There were no differences in terms of breast composition evaluation between the two readers.

A total of 59 (90.7%) abnormal findings were detected at DM+BUS by both readers. Among these, Reader 1 identified 28 lesions (47.5%) only at DM (18 of them were suspicious calcifications) and 7 (11.8%) only at BUS, while Reader 2 identified 28 lesions (47.5%) only at DM (18 of them were calcifications) and 8 (13.6%) only at BUS.

Six additional malignant lesions were identified at CEM by both readers (n = 65, 100%) (Figure 2).

In total, 14 and 15 findings were classified as benign by Reader 1 and Reader 2 at DM+BUS, respectively, and 51 and 50 (for Reader 1 and Reader 2, respectively) were classified as malignant; meanwhile, 29 and 30 findings were classified as benign by Reader 1 and Reader 2 at CEM, respectively, and 36 and 35 were classified as malignant (Figure 3 and Figure 4).

The classification of abnormal findings detected by each reader is summarized in Table 2.

Inter-reader agreement between the two readers was excellent in all cases (Cohen’s k = 0.87 for BUS and DM+BUS, 0.90 for DM, and 0.97 for CEM).

The overall diagnostic performance of CEM was superior to that of DM+BUS and of DM and BUS alone, even if to a lesser extent. In particular, CEM showed a better sensitivity (of 93.5% vs. 90.3% of DM+BUS, for both readers) and a much better specificity (of 79.4–82.4% vs. 32.4–35.5% of DM+BUS) (*p* = 0.006 for both readers). Data about the overall diagnostic performance of DM+BUS and CEM are summarized in Table 3.

ROC curves are displayed in Figure 5.

The AUC value was higher for CEM than for DM and BUS alone, and higher than for DM+BUS as well, reaching 0.865 (95% CI = 0.783–0.947) vs. 0.613 (95% CI = 0.518–0.709) for Reader 1, and 0.880 (95% CI = 0.801–0.958) vs. 0.628 (95% CI = 0.531–0.725) for Reader 2, respectively. The comparison between the AUC values was statistically significant in all the cases (*p* < 0.001 for both readers for DM+BUS vs. CEM; *p* < 0.001, for both readers for DM vs. CEM; *p* = 0.05, for Reader 1 and *p* = 0.04 for Reader 2, for BUS vs. CEM).

## 4. Discussion

The rationale of CEM, as well as CE-MRI, is that abnormal breast tissue is characterized by post-contrast enhancing.

It is nowadays established that breast MRI has the highest sensitivity in breast cancer detection (overall sensitivity of 90% [37]), but is burdened by a suboptimal specificity.

It was demonstrated that CEM has a similar diagnostic performance to breast MRI in cancer detection [24,38,39,40], with some advantages compared to breast MRI, including lower costs, with the possibility to “upgrade” existing digital mammography units [41], higher availability, and fewer contraindications. Moreover, in a recent study by Hobbs et al. [42] including 49 patients, CEM was overall preferred to breast MRI due to the faster procedure time, the greater comfort, and the lower rates of anxiety induced by the examination.

Considering the well-known limits of conventional imaging, we proposed CEM examination to patients with dense breasts and a suspicious finding (to justify increased radiation dose) as a second-level examination.

The results of our study have indicated that all parameters considered were improved in CEM compared to conventional imaging techniques alone and combined together.

In particular, CEM reached a 93.5% sensitivity, with an increase in sensitivity of 3% compared to DM+BUS and of some 13–16% compared to DM. These results are similar to those of a similar large-population-based study [31] and are comparable to the existing literature pertaining to CEM, which has reported a pooled sensitivity of 89% [43] and an improvement in sensitivity of CEM vs. DM of more than 20% in women with dense breasts [27]. Our results suggest that CEM represents a valuable tool in terms of breast cancer detection, with a possible concrete role in the screening setting.

CEM has also demonstrated a good performance in terms of the differentiation between benign and malignant lesions: 88% of lesions included in the present study were correctly classified at CEM by both readers (vs. 65% of DM+BUS). This percentage is similar to what has been previously reported by Travieso-Aja et al. [31] in a comparable large-population-based study (about 10% of lesions misclassified by CEM). Moreover, as already seen in the abovementioned similar research [31], the number of FP findings detected in CEM was higher than FN ones (only 2/8 cases in our study). However, the FP rate of CEM was much lower than that of DM+BUS (with 17 more FP findings detected by both readers at conventional imaging), with a consequent marked increase in PPV. A possible explanation for this result is that the majority of FP findings at DM+BUS corresponded actually to suspicious calcifications at DM. This evidence suggests that CEM could be of added value in case of enhancing calcifications, even if it is not currently recommended to downgrade non-enhancing suspicious calcifications to a lower BI-RADS category [44], since some authors have demonstrated that only 81% of ductal carcinomas in situ show post-contrast enhancement at CEM [45]. The second possible reason is the intrinsic high FP rate associated with high breast density at DM.

CEM was able to detect 6 additional malignant lesions that were otherwise missed at conventional imaging, enabling the correct evaluation of the extension of breast cancer and changing the therapeutic strategy for 5 patients de facto. This result supports the role of CEM as a preoperative staging tool [46].

Our study has also demonstrated the good specificity of CEM (of 79–82%), which was markedly increased compared to that of conventional techniques, both alone and together. Our results are moderately better than those found by Cheung et al. [27] in a population of women with high breast density (up to 67% specificity); however, these results are comparable to those of a recent meta-analysis of 18 studies conducted by Zhu at al. [43], which reported a pooled specificity of 84%, even if this remains highly variable across the studies (ranging from 41 to 94%) in the diagnostic setting [47,48].

Unlike one previous similar study [31], specificity and PPV of DM+BUS were lower than DM and BUS alone in this research. This unexpected result is probably due to the decision to consider the higher BI-RADS category in case of discrepancies between DM and BUS (worst case scenario), with a consequent increase in the number of wrongly assessed cases.

According to our results, CEM allowed a marked increase in NPV values compared to DM+BUS (93.1% vs. 78.6%) and DM and BUS alone, due mainly to a reduction in FN findings.

The maximum AUC value obtained in the present study (0.88) was similar to data reported in literature (0.89) [49], confirming that CEM has a high overall diagnostic performance, with a good accuracy.

Our study has some limitations: firstly, it was a single-center study including a limited number of patients; nonetheless, as far as we know, there are only three other studies [29,30,31] that have compared the diagnostic performance of CEM vs. DM and BUS combined together, and all of them included women with breasts of all densities (ACR BI-RADS a–d) while we decided to include exclusively patients with dense breasts, with a consequent obvious reduction in the number of eligible patients. In addition, a selection bias can be configured, since our sample included exclusively patients with abnormal findings at conventional imaging. Finally, the decision to include the higher BI-RADS category for DM+BUS in the analysis in case of discrepancies caused a probable bias affecting the FP rates of both tests combined together, with a consequent reduction in specificity and PPV.

In conclusion, our results, although preliminary, have demonstrated that CEM has a better diagnostic performance compared to conventional imaging techniques (alone and combined together) in patients with dense breasts, confirming the little evidence present in literature. On this basis, we believe that CEM has the potential to become a more accessible, alternative modality to CE-MRI of the breast as a second-level examination.

## Figures and Tables

**Figure 1 diagnostics-13-02520-f001:**
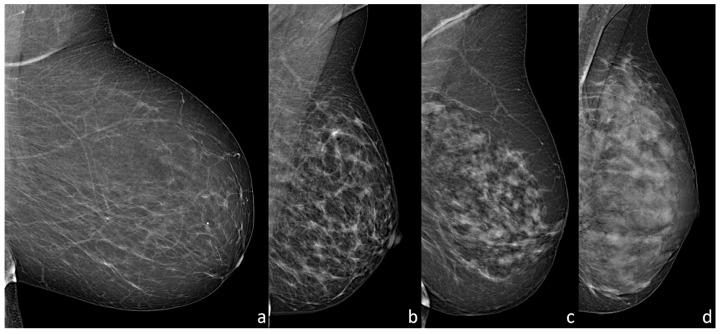
DM examinations showing different breast compositions: (**a**) almost entirely fatty breast (ACR BI-RADS a); (**b**) breast with scattered areas of fibroglandular density (ACR BI-RADS b); (**c**) heterogeneously dense breast (ACR BI-RADS c); (**d**) extremely dense breast (ACR BI-RADS d).

**Figure 2 diagnostics-13-02520-f002:**
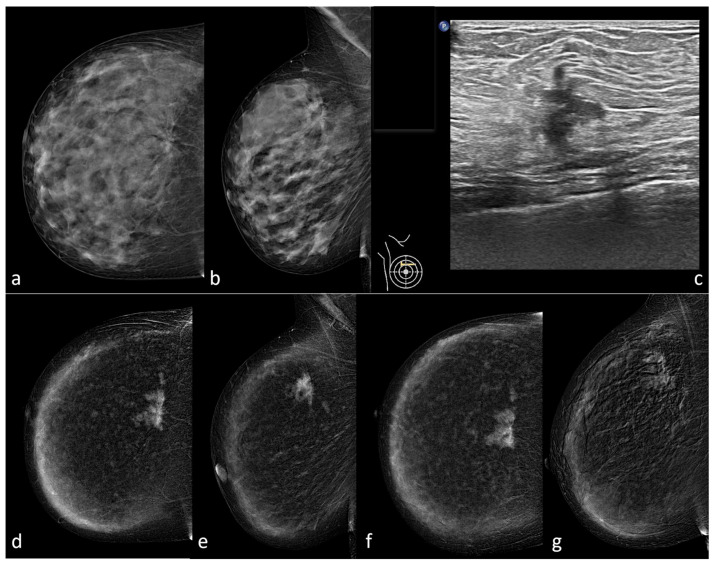
A 50-year-old woman with extremely high-density breasts (ACR BI-RADS d): CC (**a**) and MLO (**b**) views show a large architectural distortion between the upper quadrants of the right breast, with pleomorphic calcifications associated. The finding is confirmed at BUS, (**c**) which shows an irregular hypoechoic 14 mm mass with spiculated margins in the same place. CEM confirms the presence of an irregular enhancing mass with spiculated margins between the upper quadrants of the right breast and reveals the presence of an additional small enhancing mass with irregular margins in the upper-outer quadrant of the same breast. Both the lesions show fast wash-in (**d**,**e**) and fast wash-out, (**f**,**g**) and are consistent with multifocal breast cancer (BIRADS 5). The suspected diagnosis was histologically confirmed.

**Figure 3 diagnostics-13-02520-f003:**
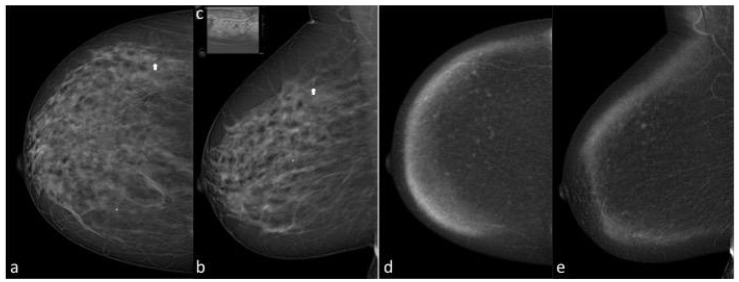
A 48-year-old asymptomatic woman with highly dense breasts (ACR BI-RADS c): CC (**a**) and MLO (**b**) views of the right breast show a cluster of calcification of new onset in the upper-outer quadrant (arrows), classified as BIRADS 4a by both readers; the finding is not confirmed at BUS (**c**). CC and MLO recombined images at CEM examination (**d**,**e**) show mild breast parenchymal enhancement, without significant post-contrast enhancement at the site of the calcifications. The patient underwent VAB. The finding was benign after histological examination.

**Figure 4 diagnostics-13-02520-f004:**
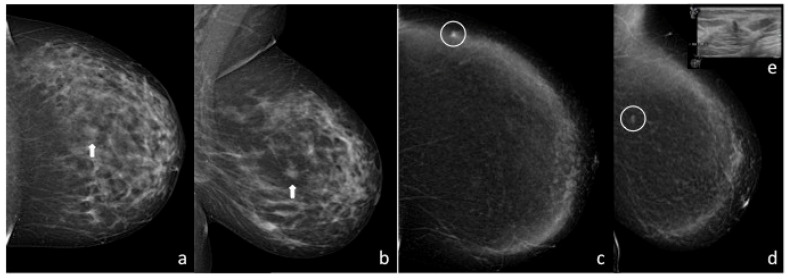
CC and MLO views of the left breast at DM (**a**,**b**) show an irregular high-density mass in the retroareolar region (arrows); the finding was not confirmed at BUS. CC and MLO recombined images at CEM examination (**c**,**d**) do not show significant post-contrast enhancement at that level but reveal a small round enhancing mass with irregular margins in the upper-outer quadrant (white circles); the finding was confirmed during a subsequent focused BUS (**e**). The patient underwent ultrasound-guided CNB that confirmed the malignant nature of the lesion.

**Figure 5 diagnostics-13-02520-f005:**
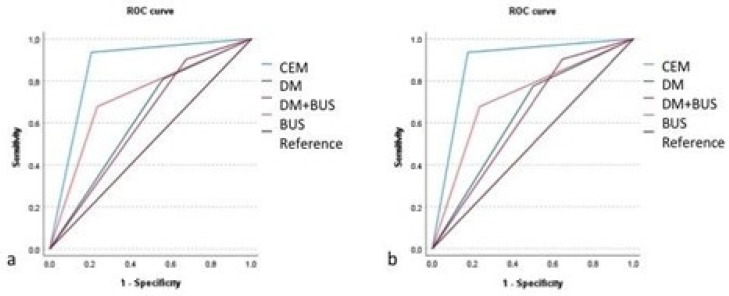
ROC curves for all the imaging techniques examined, for Reader 1 (**a**) and Reader 2 (**b**).

**Table 1 diagnostics-13-02520-t001:** CEM system technical features.

**Model**	Giotto Class with CESM	
**Manufacturer**	IMS Giotto	
**Detector**	a-Se	
**Low Energy**	Anode	W
	Filter and thickness	Al 0.7 mm
	Tube voltage range	22–35 kVp
**High Energy**	Anode	W
	Filter and thickness	Cu 0.3 mm
	Tube voltage range	40–49 kVp
**Mean glandular dose (mGy)**	1–3.1	
**Total acquisition time (s)**	10–13	

a-Se = amorphous Selenium, W = Tungsten, Al = Aluminum, Cu = Copper.

**Table 2 diagnostics-13-02520-t002:** Classification of abnormal findings detected by Reader 1 and Reader 2.

	Number of Findings	BI-RADS 1	BI-RADS 2/3	BI-RADS 4	BI-RADS 5
**Reader 1**					
DM	52 (80.0%)	7	8	29	15
BUS	31 (47.7%)	28	3	19	9
DM+BUS	59 (90.7%)	7 (10.7%)	7 (10.7%)	34 (52.3%)	17 (26.2%)
CEM	65 (100%)	23 (35.4%)	5 (7.7%)	16 (24.6%)	21(32.3%)
**Reader 2**					
DM	51 (78.5%)	8	10	31	10
BUS	31 (47.7%)	28	3	20	8
DM+BUS	59 (90.7%)	7 (10.7%)	8 (12.3%)	38 (58.5%)	12 (18.5%)
CEM	65 (100%)	24 (36.9%)	5 (7.7%)	17 (26.2%)	19 (29.2%)

**Table 3 diagnostics-13-02520-t003:** Diagnostic performance of conventional imaging techniques and CEM.

	Sensitivity	Specificity	PPV	NPV	Accuracy	AUC (95% CI)
**Reader 1**						
DM	80.6%	44.1%	56.8%	71.4%	61.5%	0.624 (0.513–0.734)
BUS	67.7%	76.5%	72.4%	72.2%	72.3%	0.721 (0.610–0.832)
DM+US	90.3%	32.4%	54.9%	78.6%	60.0%	0.613 (0.518–0.709)
CEM	93.5%	79.4%	80.6%	93.1%	86.1%	0.865 (0.783–0.947)
**Reader 2**						
DM	77.4%	50.0%	58.5%	70.8%	63.1%	0.637 (0.524–0.751)
BUS	67.7%	76.5%	72.4%	72.2%	72.3%	0.721 (0.610–0.832)
DM+BUS	90.3%	35.3%	56.0%	80.0%	61.5%	0.628 (0.531–0.725)
CEM	93.5%	82.4%	82.9%	93.3%	87.7%	0.880 (0.801–0.958)

PPV = positive predictive value, NPV = negative predictive value, AUC = area under the curve, CI = confidence interval.

## Data Availability

Data available on request.

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
