# Peer review of "Diagnostic Performance of Contrast-Enhanced Digital Mammography versus Conventional Imaging in Women with Dense Breasts"

_diagnostics, 2023, doi:10.3390/diagnostics13152520_

Round 1

Reviewer 1 Report

This article presents a comparison between the diagnostic performance of contrast-enhanced mammography (CEM) vs. digital mammography (DM) combined with breast ultrasound (BUS), in women with dense breasts. The paper is interesting, well-written, and relevant. Some minor comments can be addressed before acceptance as

1- What are the image segmentation methods that can be used in CEM compared to DM? Please discuss!

2- What are the image enhancement methods that can improve the CEM method sensitivity? Please discuss!

3- What are the factors that determine the successful detection of abnormal conditions of dense breasts using the CEM compared to DM?

4- The classification part is not very clear. Please further discuss the implemented methods and statistical results to make them easier to grasp by readers. 

5- It is recommended to publish the image dataset.

Author Response

This article presents a comparison between the diagnostic performance of contrast-enhanced mammography (CEM) vs. digital mammography (DM) combined with breast ultrasound (BUS), in women with dense breasts. The paper is interesting, well-written, and relevant.

Response: Thank you for your kind comments.

Some minor comments can be addressed before acceptance as:

  • What are the image segmentation methods that can be used in CEM compared to DM? Please discuss!

Response: Thank you for your interesting suggestion. Unfortunately, our knowledge about the topic is very limited, since we have not implemented segmentation methods (neither to analyze CEM nor DM derived images) in our clinical practice; that considered, we do not believe that discussing such methods could give an added value to our manuscript.

  • What are the image enhancement methods that can improve the CEM method sensitivity? Please discuss!

Response: As specified in the text (page 4, lines 144-146), the accuracy of CEM can be reduced by background parenchymal enhancement, exactly as contrast-enhanced breast MRI. For this reason, we decided to perform imaging of non-menopausal women in the pre-ovulatory phase of menstrual cycle (7th-14th day).

  • What are the factors that determine the successful detection of abnormal conditions of dense breasts using the CEM compared to DM?

Response: As mentioned in the introduction of our manuscript (pages 2-3), CEM is a contrast-enhanced imaging technique that combines the anatomic and morphologic information provided by a standard DM with functional information, through intravenous iodinated contrast media administration. Therefore, differently from DM, CEM is able to detect the abnormal enhancement of malignant breast lesions (due to tumoral neoangiogenesis), as well as contrast-enhanced breast MRI.

  • The classification part is not very clear. Please further discuss the implemented methods and statistical results to make them easier to grasp by readers. 

Response: Thank you for this comment. Abnormal findings were classified according to the latest edition of ACR BI-RADS lexicon. We preferred to synthesize our data in a table (in particular table 2) to avoid repetitions or the use of cumbersome sentences in the text, as suggested in the instruction for authors.

  • It is recommended to publish the image dataset.

Response: Thank you for your advice, but we believe that three different cases (figures 2-3-4), including at least two imaging techniques, could be a sufficient sample of our image dataset. As specified in our data availability statement, other data are available on request.

Reviewer 2 Report

This manuscript focused on the comparison between the diagnostic performance of CEM and DM combined with BUS, in patients with dense breast. Overall, the results are excellent and this manuscript is well-written. I just recommend authors to add descriptions of possible disadvantage of CEM as compared to DM+BUS such as cost in introduction or discussion.

Author Response

This manuscript focused on the comparison between the diagnostic performance of CEM and DM combined with BUS, in patients with dense breast. Overall, the results are excellent and this manuscript is well-written. I just recommend authors to add descriptions of possible disadvantage of CEM as compared to DM+BUS such as cost in introduction or discussion.

Response: Thank you. We really appreciate your kind comments. A few disadvantages of CEM vs. DM were already listed in the introduction; otherwise, following your suggestion, we decided to expand the introduction as follows: “The main disadvantages of CEM are the increase in radiation dose compared to DM (varying between 15-80% [19-22]) the use of iodinated contrast media, and the higher costs. Nevertheless, CEM is currently considered safe, since radiation dose - although higher - remains within acceptable levels and serious contrast agent adverse reactions are extremely rare, accounting for about 0.002% in a recent review by Zanardo at al. including more than 14000 patients [23]. For what concerns the costs of CEM, most existing digital mammography units can be implemented with dedicated filters and software to perform CEM, reducing expenses".

”.